# Clinical Utility of Endoscopic Ultrasound (EUS) and Endobronchial Ultrasound (EBUS) in the Evaluation of Mediastinal Lymphadenopathy

**DOI:** 10.3390/diagnostics15030349

**Published:** 2025-02-03

**Authors:** Dominique Béchade

**Affiliations:** Department of Medical Oncology, Institut Bergonié, Comprehensive Cancer Center, 33076 Bordeaux, France; dominique.bechade94@orange.fr

**Keywords:** endosonography, mediastinal lymphadenopathy, mediastinal malignancies, mediastinal infections, sarcoidosis

## Abstract

In recent years, the combination of endobronchial ultrasound and endoscopic ultrasound has enabled “medical exploration” of the mediastinum for the study of mediastinal lymphadenopathies. These techniques are particularly important for the diagnosis and staging of lung cancers. Progress has been made with the availability of new-generation cutting needles for endoscopic ultrasound and new cryobiopsy needles for endobronchial ultrasound to improve the quality of samples.

## 1. Introduction

Endoscopic ultrasound (EUS) and endobronchial ultrasound (EBUS) are minimally invasive procedures that can be used to collect and characterise tissue acquisition (TA) of mediastinal lymphadenopathies. The aim of this clarification is to summarise the literature published in recent years on the role of these two endoscopic techniques in the diagnostic management and staging of the main pathologies associated with mediastinal lymphadenopathies.

## 2. Exploration of the Mediastinum with EBUS and EUS

The diagnostic performance of each available technique depends on the location of the lymphadenopathy in one of the mediastinal stations (Figure 1) [1].

### 2.1. EBUS, EBUS-Guided Fine Transbronchial Needle Aspiration (EBUS-TBNA) and EBUS-Guided Mediastinal Cryobiopsy (cryoEBUS)

EBUS identifies pre-tracheal and hilar adenopathy located in anterior stations 2, 4 and 7 and, unlike EUS, explores adenopathy in stations 10, 11 and 12 (Figure 2) [2].

The primary indication for EBUS-TBNA is the diagnosis and the evaluation of resectability or restaging following chemotherapy or radiochemotherapy for non-small cell lung cancer (NSCLC) [3]. It is also used to diagnose pulmonary and/or mediastinal masses in a known or unknown context of extrathoracic neoplasia [4,5]. EBUS-TBNA can be used to sample adenopathies measuring up to 5 mm in the anterior and superior mediastinum and pulmonary hilar region [6]. The rate of severe complications, such as pneumothorax or respiratory failure, is estimated to be 0.05% [6].

Concerning the needles used with EBUS-TBNA, some series have showed that 19G needles are very suitable for obtaining large histologic particles in lymph nodes and for molecular pathology [7]. The cell area obtained via the 19G needles appear significantly larger than that obtained with the 21G needles, especially for lymphoproliferative disorders [8].

Nevertheless, newer needle systems such as EBUS-guided mediastinal cryobiopsy (cryoEBUS) [9,10,11] have been developed to maximise the diagnostic yield of mediastinal lymphadenopathy by providing additional tissue for the diagnosis and molecular evaluation of some lung cancers [12] and mediastinal lymphomatous pathologies [11]. Initially proposed for interstitial lung diseases, cryoEBUS is based on cooling, crystallisation and subsequent collection of tissue in larger quantities. By extension, it is now used to diagnose mediastinal lymph nodes [13]. A randomised controlled trial compared cryoEBUS to EBUS-TBNA in 197 patients with mediastinal lymphadenopathies: the diagnostic yield was higher with cryoEBUS compared to EBUS-TBNA (91.8% vs. 79.9%, *p* = 0.001). The sensitivity of cryoEBUS was higher for lymphoma and for benign lesions (sarcoidosis and tuberculosis) [14]. A recent meta-analysis showed that cryoEBUS improves diagnostic yield over EBUS-TBNA in benign and possibly lymphoproliferative diseases, but less in lung cancer [15]. Similar results were already found in other series [16]. Complications of cryoEBUS occur in 38% of cases [9,15]: most are self-limited bleeding during the procedure not requiring any intervention (36%); significant bleeding occur in 0.7% of cases. Pneumothorax is reported in 0.4% of cases (0.5% for EBUS-TBNA). Concerning molecular pathology, a retrospective study including 37 patients showed that cryoEBUS is more effective in obtaining adequate samples for next-generation sequencing (NGS) analysis compared to EBUS-TBNA [17]. When cryoEBUS is not available, the molecular pathology must be performed with a 19G needle [7]. In this case, the slow-pull technique instead of succion technique should be used, especially for programmed death ligand-1 (PD-L1) testing in NSCLC [18] and for the NGS testing in NSCLC [19]. Elastography can also be used to differentiate between pathological and non-pathological lymphadenopathies and can therefore help in selecting lymphadenopathies for sampling [20].

### 2.2. EUS, EUS Fine-Needle Aspiration (EUS-FNA) and Fine-Needle Biopsy (EUS-FNB)

EUS is used to explore subcarinal lymphadenopathies located in the posterior and inferior mediastinum and in the aortopulmonary window. Lymphadenopathy at stations 5, 8 and 9 and posterior lymphadenopathy at station 7 are accessible to exploration by EUS (Figure 2).

A 19G or a 22G needle is used for EUS-FNA [21]. The samples are used to perform spread cytology, stained with Diff-Quick or May–Grünwald–Giemsa, combined with liquid medium cytology for complementary cytochemical and immunocytochemical techniques. Alcohol or buffered formalin fixatives allow for good morphology and additional studies. The mediastinal lymphadenopathies can now be punctured with new-generation cutting needles: 20G side-fenestrated forward-facing bevel needles (Procore^TM^, Cook Medical, Bloomington, IN, USA) or the 22G Franseen needle (Acquire^TM^, Boston Scientific, Burlington, MA, USA), which allow real biopsies to be taken that can be retrieved for paraffin inclusion and histological examination (cell-block, standard histology and touch imprint cytology) [22,23]. Other end-cutting needles are available (Sharkcore^TM^ (Medtronic, Minneapolis, MN, USA) and Trident^TM^ (Micro-Tech Endoscopy, Ann Arbor, MI, USA)) [24]. The 19G needles and the new-generation cutting needles allow for molecular pathology [22,25]. If lymphoma is suspected, the preferred phenotyping method is cell-block or flow cytometry; if lymph node metastases are suspected, the diagnosis can be confirmed by spreads and liquid cytology. Cytological and histological examinations, on the other hand, are less effective for the identification of infectious agents; if tuberculosis is suspected, the samples must be spread on slides for Ziehl–Nielsen staining.

## 3. Result of Combined Mediastinal Exploration Techniques According to Clinical Circumstances

In the published series, mediastinal lymphadenopathies were primarily explored in the context of lung cancer (LC). We present the results of the main studies, differentiating between the LC and non-LC contexts.

### 3.1. In the Context of LC

#### 3.1.1. EUS in the Evaluation of LC

Most of the series published were carried out in patients with suspected LC and lymph nodes measuring more than 10 mm along their minor axis [2,26,27]. In these studies, the sensitivity of EUS-FNA/FNB was 90% overall and specificity was 100% [2]. A broad meta-analysis showed a sensitivity of 88% and a specificity of 96% [26]. This study alone verified the positivity of EUS-FNA compared to the result of surgical lymph node TA, concluding at a 2% false positive rate [26]. Currently, last-generation needles and tools (contrast-enhanced EUS and elastography) increase the accuracy of lung cancer TA and staging [28].

#### 3.1.2. EBUS in the Evaluation of LC

The older series reported sensitivities ranging from 79% to 95% [2] and a specificity of 100%, but positive samples have not been verified. Since then, several studies have compared the diagnostic performance of EBUS-TBNA with that of mediastinoscopy, showing a sensitivity of 55.1%, a specificity of 100%, a positive predictive value (PPV) of 100%, a negative predictive value (NPV) of 95% and a diagnostic accuracy of 81.2% [29]. More recently, Torre et al. [3] reported their experience of 270 EBUS-TBNAs performed for the diagnosis and staging of lung cancer: the average diagnostic yield was 97.12% for the diagnosis of primary lung cancer. The sensitivity, specificity, PPV and NPV for the diagnosis of malignancy were 96%, 100%, 100% and 76%, respectively. Out of this series of 270 patients with a definitive diagnosis of cancer, molecular testing was carried out on 142 patients, looking for epidermal growth factor receptor (EGFR), anaplastic lymphoma kinase (ALK), c-ros oncogene 1 (ROS1) and PD-L1 mutations. These were found in 19 patients (16.9%), with a diagnostic adequacy of 78.87% (112 out of 142 patients).

#### 3.1.3. Results of Combining These Two Endoscopic Ultrasound Techniques

In 2005, the concept of a combined “medical” approach involving EUS and EBUS was proposed as a means of mediastinal lymph node evaluation in the management of LC [30]. In a series of 33 patients with known or suspected LC, Vilmann et al. [30] showed that the diagnostic accuracy of the endoscopic combination was 100% for LC lymph node staging, whereas those of EUS-FNA alone and EBUS-TBNA alone were 86% and 89%, respectively; the 93% sensitivity of the endoscopic combination was superior to that of each isolated procedure and the 97% NPV approached that of thoracotomy with mediastinal lymph node dissection.

In the case of apparently resectable LC without mediastinal adenopathy visible on computed tomography (CT) and/or positron emission tomography (PET), the prevalence of mediastinal metastases remains between 20% and 25% [8], illustrating the value of the EUS/EBUS combination in this situation: in patients with suspected LC without a suspicious mediastinal lymph node by CT, Szlubowski et al. [31] showed, in a series of 120 patients, that the sensitivity (68%) and NPV (91%) of combined fine-needle aspiration by EUS and EBUS were significantly higher than those of EUS-FNA alone and EBUS-TBNA alone. In this study, the prevalence of malignant mediastinal metastases was 22%.

Wallace et al. [32] studied the results of the EUS/EBUS combination in patients with suspected LC investigated by CT and PET: when CT and/or PET showed supracentimetric mediastinal adenopathy, the NPV of the EUS/EBUS combination was 100%. When CT and PET showed no mediastinal adenopathy, the NPV dropped to 94%. The prevalence of malignancy was 20% in the CT and PET-negative subgroup, documented by cytological diagnosis or surgical confirmation.

In a series of 241 patients with potentially resectable NSCLC, the ASTER trial (multicentre randomised controlled trial) [33] compared surgical staging alone with combined endoscopic ultrasound staging followed by surgery if no lymph node metastases were diagnosed by endoscopic ultrasound. Sensitivity was 79% for mediastinoscopy staging alone. Although this study showed that mediastinoscopy, performed after a negative combined endoscopic ultrasound procedure, increased lymph node metastasis detection sensitivity from 85% to 94%, the authors concluded that this strategy should certainly be reserved for a sub-group of patients yet to be identified, as eleven mediastinoscopy procedures were necessary to detect one case of single-level N2 disease. The ASTER trial has raised the question of how to identify false-negative endosonography cases in order to abandon additional surgical staging.

In 2011, Ohnishi et al. [34] showed that the diagnostic precision of the endoscopic ultrasound combination was higher than that of PET (90% vs. 73.6%, respectively); the sensitivity, specificity, PPV and NPV were 71.8%, 100%, 100% and 86.6% for the endoscopic ultrasound combination and 47.4%, 87.5%, 66.7% and 75.9% for PET, respectively. Given these results illustrating the therapeutic impact of the endoscopic ultrasound combination, the authors propose a decision tree integrating it into the management strategy for potentially resectable LC after an extension assessment by CT and PET [34]. Mediastinoscopy can still be used as a second-line procedure in the case of N0/1 stagings provided by the endoscopic ultrasound combination.

Due to the good diagnostic accuracy of combined endosonography, the MEDIASTrial study group published two multicentre randomised trials to demonstrate that mediastinoscopy after negative systemic endosonography can be omitted in patients with resectable NSCLC [35,36]. They demonstrated that mediastinoscopy reduced the unforeseen N2 (uN2) rate by only 1.03%, with the disadvantage of a 10-day delay for lung tumour resection, a morbidity of 6.3% and a mortality of 0.6% [36]. All these data were also found in a meta-analysis which included 42 studies and a total of 3248 patients, showing that the rate of uN2s after negative endosonography was similar in patients who underwent surgical resection with or without prior staging mediastinoscopy [37].

Consequently, a new technique using a convex probe–endobronchial ultrasound scope through the oesophagus (endoscopic ultrasound with bronchoscope—EUS-B) has emerged as a preferable unique procedure for some pulmonologists. The results of this original endoscopic approach, using a single ultrasound bronchoscope to simultaneously perform endobronchial ultrasound and endoscopic ultrasound, have been published [38]: in a series of 150 patients with suspected NSCLC, Herth et al. [39] showed that the sensitivity and NPV of EBUS-TBNA, EUS-B-TA and the combined approach were 91% and 92%, 89% and 82%, and 96% and 96%, respectively.

In a prospective, multicentre, single-arm international study (SCORE study), Crombag et al. [40] reported that the combined use of EUS-B following a systematic EBUS procedure (EBUS/EUS-B) increased sensibility for detecting mediastinal metastases by 5% (from 77% to 82%). A review and meta-analysis even showed that the addition of EUS-B to EBUS leads to a 12% gain in the detection of mediastinal nodal metastases [41].

Oki et al. [42] published a randomised study to assess the appropriateness of EBUS alone compared to EBUS/EUS-B. They found that EBUS/EUS-B exhibited a higher sensitivity (4.3%) than EBUS did alone. Probably because EBUS and EUS-B explore some of the same stations, EBUS/EUS-B should be preferred to EBUS alone because EBUS/EUS-B achieves optimal sensitivity by expanding the range of evaluable lymph nodes [43].

Recently, 141 patients with suspected malignant mediastinal lymph nodes with known or suspected lung cancer, as identified by PET-CT, underwent systematic combined EBUS-TBNA and EUS-TA to evaluate the respective contribution of separate and combined procedures [44]. Unlike most series, this study underlines the importance of each technique and shows the significant role played by EUS-TA. One of the most important differences was the use of FNB needles in the majority of patients. The sensitivity for malignancy diagnosis of EBUS-TBNA, EUS-TA and combined EBUS-TBNA/EUS-TA were 75%, 87% and 93%, respectively. Staging was also significantly improved by the combined approach with a statistically significant difference in sensitivity of 12% between EBUS-TBNA and the combined approach and between EUS-TA and the combined approach.

This study should encourage the two types of equipment to be grouped together on the same technical platform [28], as these two endoscopic techniques are performed by different operators (gastroenterologists and pulmonologists). When this is not possible, the use of combined EBUS/EUS-B eliminates the need for repeated anaesthesia procedures or travel from one hospital to another. Moreover, we now have precise guidelines on whether the two techniques should be used in combination [45]. All situations are discussed, and specific management is proposed.

#### 3.1.4. Re-Evaluation After Induction Treatment

Mediastinal lymph node re-evaluation after induction treatment is important to identify patients likely to benefit from surgical treatment. The most important study focused on EBUS-TBNA [46], which was compared with the results of thoracotomy in a series of 124 patients with initial-stage IIIA-N2 non-small-cell LC: 28 EBUS-TBNA false negatives were observed, showing that the sensitivity of the technique for the detection of residual N2 mediastinal disease after induction chemotherapy was 76%, with 100% specificity and 77% diagnostic accuracy. The NPV and PPV were 20% and 100%, respectively; this low NPV indicates that, in the context of mediastinal re-evaluation, EBUS-TBNA is a good technique for confirming but not excluding the presence of mediastinal metastases requiring surgical staging when the EBUS-TBNA results are negative for malignancy.

### 3.2. Outside the Lung Cancer Context

#### 3.2.1. Malignancies

##### Lymph Node Metastases of Extrapulmonary Cancers

Many cancers cause mediastinal lymph node metastases: oesophageal, breast (Figure 3), ENT, stomach, kidney, pancreatic and gynaecological cancers [47]. When treating or monitoring cancer, the wider use of PET significantly increases the rate of screening for these lymph nodes. However, the high sensitivity of PET can lead to diagnostic interpretation difficulties affecting the therapeutic strategy. In the absence of a history of neoplasia, mediastinal adenopathy is thought to be associated with extrapulmonary cancer in 15% of cases [48]; if there is a history of neoplasia, it is thought to reflect either the recurrence of a thoracic cancer (LC or oesophageal cancer), the recurrence of an extrathoracic cancer or the appearance of a second cancer in 17% of cases (LC or non-Hodgkin’s lymphoma) [48]. The therapeutic impact of endoscopic ultrasound-guided TA, as a replacement for invasive surgical procedures, has already been shown by our team [49].

##### Lymphomas

The EUS/EBUS combination is useful in this setting, especially in patients with a history of lymphoma in whom recurrence is suspected [50,51]. Samples must be subjected to flow cytometry immunophenotyping.

Although the impact of the EUS/EBUS combination appears effective for the diagnosis of mediastinal lymphoma, its sensitivity is only 73–80% [50,52]. Thus, obtaining a TA by a surgical procedure is desirable if the sample obtained by the combined procedure is negative [51]. Similarly, prognostic criteria for certain lymphoma subtypes can only be recognised from a tissue architecture, not always allowed by fine-needle aspiration. However, EUS-FNB allows for true histological analysis, especially when performed with a large-gauge needle. EUS-FNB, when coupled with flow cytometry, is accurate in the diagnosis and classification of diffuse large B cell lymphoma. The diagnostic accuracy of this technique is lower for low-grade lymphomas, Hodgkin’s disease and T cell lymphomas [53]. The near-zero morbidity associated with this procedure means that the number of intra-lesion passages is virtually unlimited, allowing for sufficient material to be collected for all the complementary techniques (histology, flow cytometry and cytogenetics in particular). Thus, EUS-FNB is increasingly used for the initial diagnosis and recurrence of B cell lymphomas but remains more debated for low-grade lymphomas, Hodgkin lymphomas and T cell lymphomas [52]. The availability of needles with a core trap cut-out and an inverted bevel, designed for the collection of histological grade material (Procore^TM^ needles), allow for better quality analysis [54]. As mentioned above, the advances brought about by these new-generation cutting needles in EUS are associated with new diagnostic performances of cryoEBUS for lymphoproliferative diseases [11,15].

#### 3.2.2. Benign Conditions

##### Sarcoidosis

A diagnosis of sarcoidosis is based on a combination of clinical, biological, radiological and sometimes histological factors. PET scans also form part of the diagnostic criteria, and EBUS [5] or EUS/EBUS combination is generally used to rule out lymph node metastases in a neoplastic context [55,56] (Figure 4). Granulomas without caseous necrosis are sometimes difficult to diagnose on samples obtained by fine-needle aspiration; diagnostic efficiency can be improved by the use of larger-gauge needles such as 19G or by the use of Procore^TM^ needles [49,55]. The different series show a sensitivity ranging from 86% to 94% and a specificity of 96% [55].

##### Infections

Tuberculosis or other infections can be diagnosed in samples obtained by EUS and/or EBUS [57]. When diffuse polyadenopathy raises the suspicion of systemic infection, samples obtained by EUS or EBUS should be specifically stained and cultured, including fungal cultures. Sample quality is improved by the use of Procore^TM^ needles [22].

## 4. Conclusions

In recent years, mediastinal adenopathy has been assessed using endoscopic techniques. New-generation cutting needles for endoscopic ultrasound and new cryobiopsy needles for endobronchial ultrasound improve the quality of samples. Combining EUS and EBUS offers a less-invasive diagnostic approach with a high therapeutic impact, particularly in the context of LC extension assessment but also in the context of other neoplasias. The combined EBUS-TBNA and EUS-FNA/FNB or EUS-B procedure for mediastinal lymph nodes allows for interpretable specimens to be obtained in more than 95% of cases, with a specificity close to 100% and a sensitivity ranging from 88% to 96%.

## Figures and Tables

**Figure 1 diagnostics-15-00349-f001:**
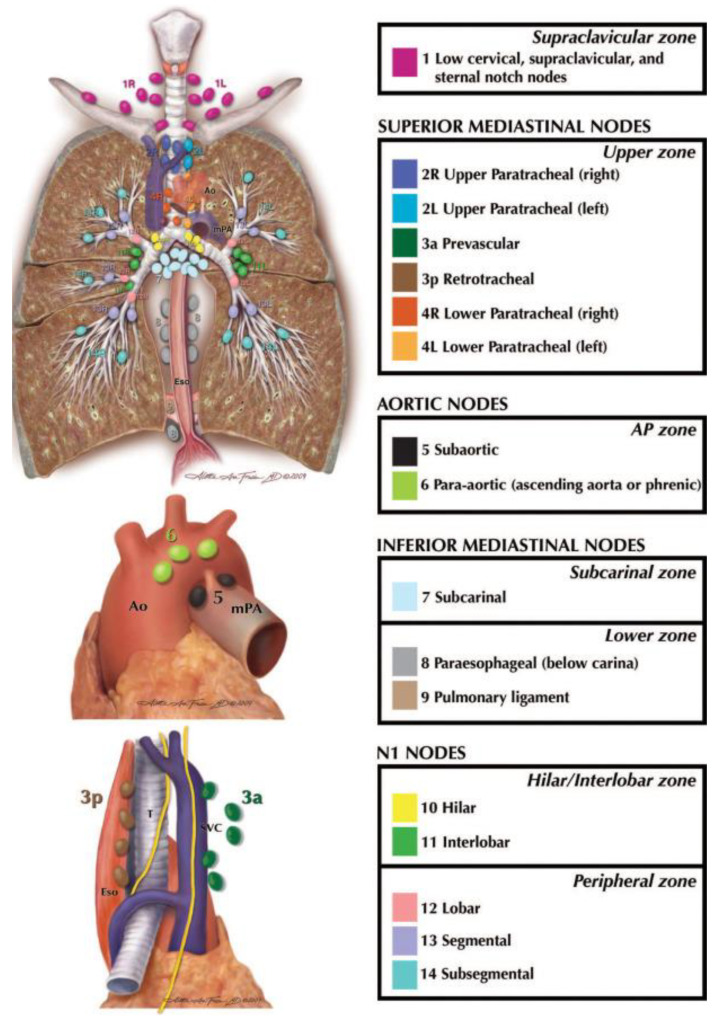
International Association for the Study of Lung Cancer (IASLC) classification (mPA: mean pulmonary artery; T: trachea; SVC: superior vena cava; Eso: oesophagus).

**Figure 2 diagnostics-15-00349-f002:**
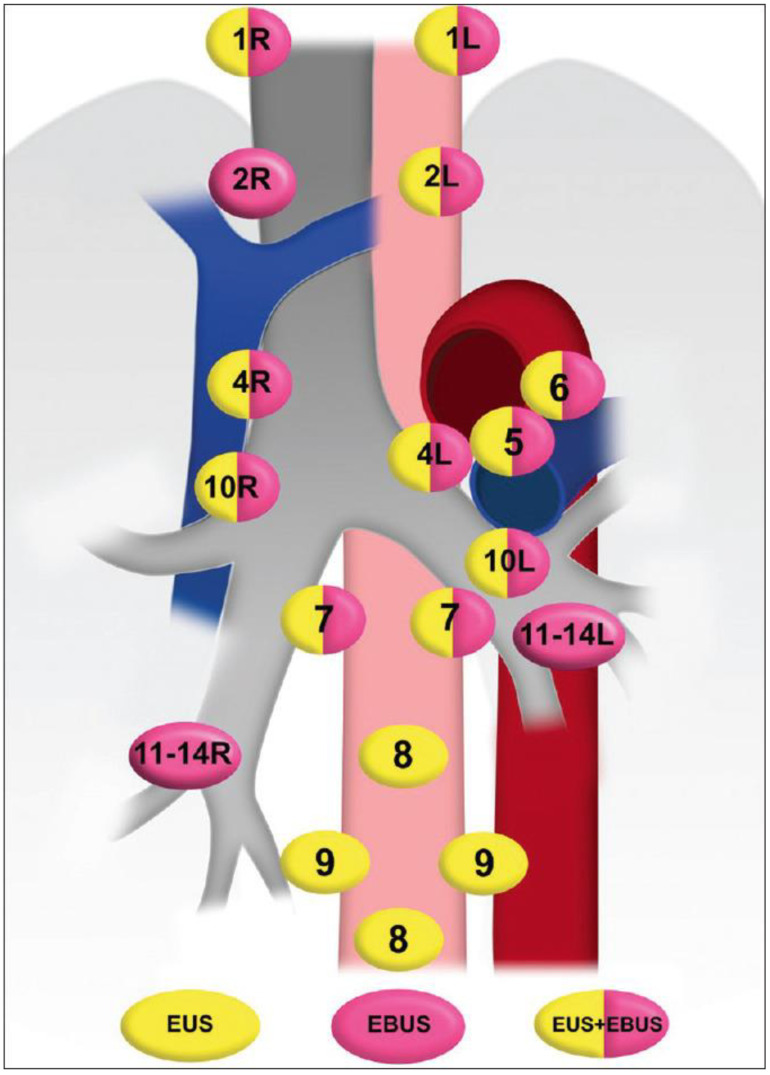
Stations of the mediastinum showed by endoscopic ultrasound (EUS), endobronchial ultrasound (EBUS) and combination EUS/EBUS.

**Figure 3 diagnostics-15-00349-f003:**
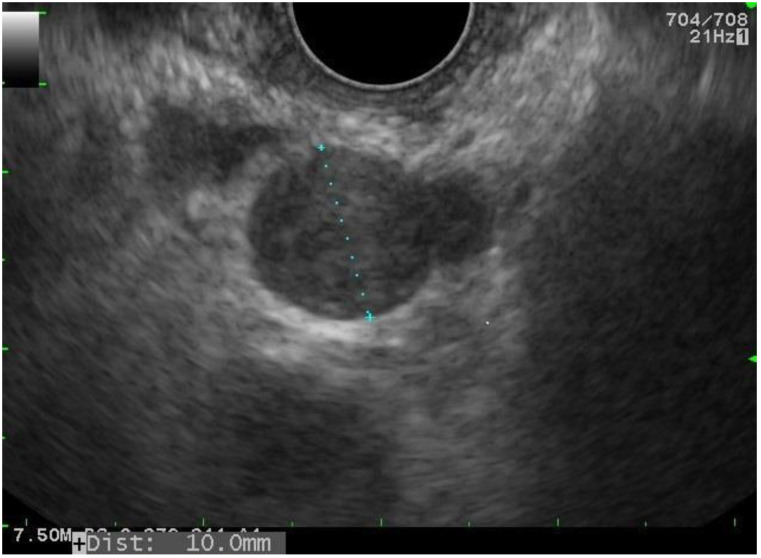
Mediastinal lymph node metastasis of breast cancer: lumpy, homogeneous hypoechogenic mass, associated with hyperechogenic reinforcement.

**Figure 4 diagnostics-15-00349-f004:**
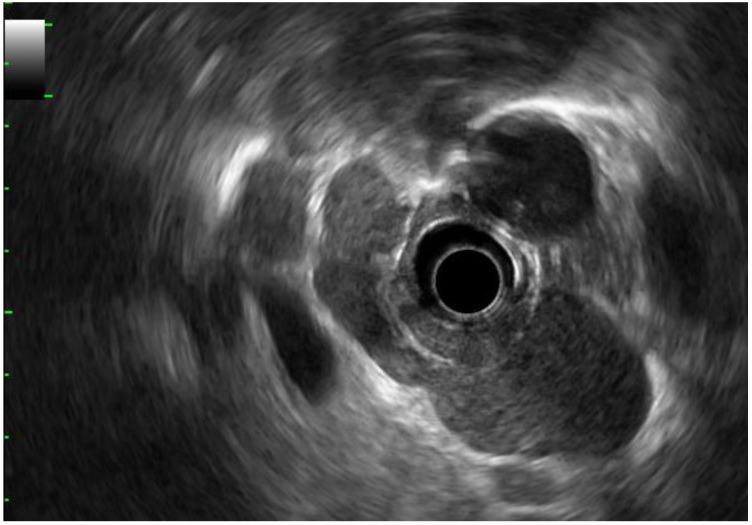
Voluminous, non-compressive, hypoechogenic and homogenous sarcoid mediastinal adenopathies, with clearly defined margins.

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
