# Peer review of "Clinical Utility of Endoscopic Ultrasound (EUS) and Endobronchial Ultrasound (EBUS) in the Evaluation of Mediastinal Lymphadenopathy"

_diagnostics, 2025, doi:10.3390/diagnostics15030349_

Round 1
Reviewer 1 Report
Comments and Suggestions for Authors
The combined approach of EBUS und EUS has been described in detail but the most important papers are not cited. Perhaps this is the most important issue.
It is also not clear to me what is new in this paper?
It is more a book chapter than a review paper.
Quite a few issues should be taken into account in more detail.
General remarks
The authors describe the extremely interesting and current topic of mediastinal lymph node assessment and material collection by EBUS and transesophageal EUS. The manuscript is very readable and interesting. However, it is noticeable that most of the references are older. These have of course all contributed greatly to the evaluation of the methods. However, one wonders whether there are also newer aspects. There are 56 references in total, but after 2010, i.e. from the last 14 years, only three are from 2011, one from 2012 and one from 2022. For example, how do the methods described reflect the requirements of current guidelines on lung cancer?
The basis of the therapy is the collection of material for molecular pathology. How is this ensured by the methods and needle systems described? What data is available?
The work is well worth reading and provides good basics on EUS and EBUS in mediastinal lymph node staging and diagnostics. However, it should be checked whether there are any current data that are worth including.
More specific remarks
Title
Please consider writing “lymphadenopathy” instead of “adenopathy” in the title.
Abstract
The abstract summarizes the study in an informative manner.
Keywords
Please consider writing “lymphadenopathy” instead of “adenopathy” in the keywords.
Introduction
The introduction provides an informative background to the work.
This is primarily about EBUS and EUS. I would therefore not start with the sentence:
Line 15-16: “Performing mediastinal adenopathy biopsy under CT is difficult due to the proximity of major vascular structures.” CT is not actually an indication for biopsy of mediastinal lymph nodes. You could put the sentence at the end of the introduction.
Review:
Line 43: “The two paraclinical circumstances for the discovery of mediastinal adenopathy” - Is this a subheading?
Line 74-75: “Fine needle aspiration techniques” – Is this a subheading?
Line 76-78: Transthoracic biopsy: CT-supported biopsy is mentioned here. I would explain that these are not lymph nodes but pleural and pulmonary lesions that are biopsied. Wouldn't transthoracic ultrasound-guided sampling of pleural and thoracic wall lesions also be worth mentioning in this context? One could even mention that CEUS-guided sampling allows the differentiation of necrosis and vital tumor tissue. One sentence with references is sufficient.
Line 85: The word “adenomegaly” is unusual. Can it be replaced? “Enlarged lymphnodes” or something else.
Subheading 4.5 – line 88-95: Are there newer needle systems? How do you rate TB cryobiopsy? Is it worth mentioning - positive and negative aspects? Tumor seeding?
Subheading 4.6 – line 96-117: Why is the 19 G needle not mentioned here? It is very suitable for obtaining large histology particles in lymph nodes and for molecular pathology. The 19 G needle will be mentioned at a later stage in the diagnosis of lymphoma diseases.
What about TB-EBUS sampling as well as EUS-guided sampling with material for molecular pathology? What are the actual study results? What material has to be supplied, what are the results for the different needle systems for MolPath? Could you insert some explanations and data.
You describe the criteria for pathological lymph nodes for EBUS and then for EUS. Shouldn't they be identical? You quote the size cut of 8.3 mm. What is the significance of the different characteristics? In my experience, any round, hypoechoic lymph node near a tumor is tumor-suspect, regardless of size.
I would still describe that the central hyperechoic reflex that you describe as a criterion for benignity corresponds to the central lymph node hilus. Then I would write a few words about the “vascular pattern” of lymph nodes. It doesn't have to be detailed. In malignant lymph nodes, the normal vascular architecture is also destroyed and the lymph node hilus can no longer be demarcated.
And what about strain elastography? What role does it play in the assessment of pathological versus non-pathological lymph nodes, can elastography be helpful in the selection of a lymph node for sampling?
Line 206 and following:
Line 252: The word “enchoendoscope” is not correct. You probably mean EBUS-EUS probes for transbronchial and transesophageal EUS?
Conclusions
The Conclusions describe that EUS and EBUS combined can visualize the lymph node stations as far as possible and mediastinoscopy can be avoided with a few exceptions.
In the Conclusions, one could again formulate in which patients it is indicated to use both endosonographic methods. If, for example, transesophageal EUS detects tumor lymph nodes that rule out primary operability, EBUS is no longer required in these patients, as the procedure does not change. But conversely, should all patients who appear to be primarily operable be examined using both methods? What do the guidelines say?
In line 337 you use the word “non-invasive”. I would formulate it as less invasive, since EBUS and EUS are also invasive in a certain sense.
Author Response
Thank you for all your comments. All modifications are underlined.
General remarks
Many recent references have been added.
The new needles available for molecular pathology have been described.
New current data have been included.
More specific remarks
Title: “lymphadenopathy” was written instead of “adenopathy”.
Keywords: “lymphadenopathy” was written instead of “adenopathy”.
Introduction: the first sentence has been deleted and repeated later in the text.
Review:
Line 43 and line 74-75 have been modified.
Line 76-78: we have taken your comments into account and added a reference.
Line 85: we have replaced “adenomegaly” with “enlarged lymphnodes”.
Subheading 4.5 – line 88-95: the newer needle systems and cryoEBUS have been described. There positive and negative aspects have been specified. We did not find any reference about tumor seeding.
Subheading 4.6 – line 96-117: we have specified the place of 19G needles, especially for lymphoma disorders.
The various sampling methods for EBUS as well as EUS for molecular pathology have been specified.
We have deleted the comments concerning size criteria that may be predictive of pathology and we share your opinion.
A reference to elastography has been added.
Line 206: the word “echoendoscope” has been deleted and replaced by “EBUS-EUS probe”.
We have referred to the guidelines on the indications for the EUS/EBUS combination.
Line 337: the word “non-invasive” has been deleted and replaced by “less invasive”.
Reviewer 2 Report
Comments and Suggestions for Authors
This review reports the current possibilities of endoscopic tissue acquisition of adenopathy of the mediastinum. Below are my comments to improve the manuscript:
1) The abstract should be improved by reporting the strength of this review and why the readers should read it.
2) Add some figures of esophageal and endobronchial ultrasound showing the LN stations
3) EUS-FNB. Besides Procore and Acquire, other end-cutting needles are available (SharkCore and Trident). Please also mention these needles and quote this reference (PMID 31010744)
4) A section dedicated to specimen handling should be added. The authors cited the cytoblock, but no reference is available. However, cytoblock should be "cell-block," "standard histology," and "touch imprint cytology" should be mentioned. Here, I suggest some references that could be quoted (PMID 26063033).
5) Figure 2 is not clear. Please refer to the colors of different mediastinal stations.
6) A table to summarize the results of the study exploring the combination of EUS and EBUS should be added
7) Line 304, only EUS-FNB, not FNA, allows obtaining histological samples
8) Please add a section on contraindications and adverse events related to EUS and EBUS tissue acquisition
9) A native speaker should edit English.
Minor:
1) Use "endoscopic ultrasound" instead of "echoendoscopy" and "tissue acquisition" instead of "biopsy"
2) Write "computed tomography" before the abbreviation CT at line 15, introduction
Comments on the Quality of English LanguageEnglish language should be revised
Author Response
Thank you for all your comments. All modifications are underlined.
1)The abstract have been improved to reporting the strength of the review, especially the use of the new needle systems for EUS and EBUS and all their applications accordingly.
2)We have added Figure 3 to show the LN explored by EUS and EBUS and combination of EUS and EBUS.
3)We mentioned Sharkcore and Trident. The reference PMID 31010744 has been added.
4)The reference PMID 26063033 has been added and we have mentioned “cell-block”, “standard histology” and “touch imprint cytology”.
5)Figure 2: we refered to the colors of different mediastinal stations.
6)We found it difficult to summarize the results of studies exploring the combination of EUS and EBUS in a single table.
7)Line 304: EUS-“FNA” has deleted.
8)The side effects of EUS and EBUS were discussed throughout the text of the review
9)We used “endoscopic ultrasound” instead of “echoendoscopy” and “tissue acquisition” instead of “biopsy”.
10)We wrote “computed tomography” at line 15.
Round 2
Reviewer 1 Report
Comments and Suggestions for Authors
The authors responded to some but not to all points in a reasonable fashion.
The combined approach of EBUS und EUS has been described in detail in the literature but the most important papers are still not cited.
Please indicate in bullet points what is new in this paper?
Please comment if this paper is more a book chapter than a review paper.
Author Response
We have tried to include in the text the most important papers concerning the combined EBUS and EUS approach.
The points that seem new to us are dominated by technical advances concerning the new needles used in EBUS (cryobiopsies) and EUS (fine needle biopsies), as well as new techniques such as elastography to increase diagnostic performance.
This paper is intended to form part of a general review on "advances in ultrasound". For this reason, given to the wide range of different topics covered in this review, we have included some general notions of anatomy for example. This is why it resembles a book chapter.
Reviewer 2 Report
Comments and Suggestions for Authors
I have no further comments
Author Response
We had no comments from reviewer 2
Round 3
Reviewer 1 Report
Comments and Suggestions for Authors
I am not totally sure if the author takes the comments serious, e.g., see in the abstract still adenopathies are mentioned "The endosonographic approach of mediastinal adenopathies has been transformed". ...
What is really new?
Author Response
I completely agree with the reviewer: there is nothing really new in this paper.
Consequently, this review of the literature has been considerably reduced and restructured, with the deletion of several chapters (for example, on anatomy, the place of surgical means, etc...)
In addition, all sentences such as the one quoted by the reviewer in the abstract have been deleted and modified.
Round 4
Reviewer 1 Report
Comments and Suggestions for Authors
No more comments